# DASH Dietary Pattern and Cardiometabolic Outcomes: An Umbrella Review of Systematic Reviews and Meta-Analyses

**DOI:** 10.3390/nu11020338

**Published:** 2019-02-05

**Authors:** Laura Chiavaroli, Effie Viguiliouk, Stephanie K Nishi, Sonia Blanco Mejia, Dario Rahelić, Hana Kahleová, Jordi Salas-Salvadó, Cyril WC Kendall, John L Sievenpiper

**Affiliations:** 1Toronto 3D Knowledge Synthesis and Clinical Trials Unit, Clinical Nutrition and Risk Factor Modification Center, St. Michael’s Hospital, Toronto, ON M5C 2T2, Canada; laura.chiavaroli@alumni.utoronto.ca (L.C.); effie.viguiliouk@mail.utoronto.ca (E.V.); s.nishi@mail.utoronto.ca (S.K.N.); sonia.blancomejia@mail.utoronto.ca (S.B.M.); cyril.kendall@utoronto.ca (C.WCK.); 2Department of Nutritional Sciences, Faculty of Medicine, University of Toronto, Toronto, ON M5S 1A8, Canada; 3Department of Endocrinology, Diabetes and Clinical Pharmacology, Dubrava University Hospital, 10000 Zagreb, Croatia; dario.rahelic@gmail.com; 4School of Medicine, University of Zagreb, 10000 Zagreb, Croatia; 5Institute for Clinical and Experimental Medicine, Diabetes Centre, 140 21 Prague, Czech Republic; hana.kahleova@gmail.com; 6Physicians Committee for Responsible Medicine, Washington, DC 20016-4131, USA; 7CIBER Fisiopatología de la Obesidad y Nutrición (CIBERObn), Instituto de Salud Carlos III, Madrid 28029, Spain; jordi.salas@urv.cat; 8Human Nutrition Unit, Biochemistry and Biotechnology Department, Hospital Universitari de Sant Joan de Reus, IISPV, Universitat Rovira i Virgili, Reus 43201, Spain; 9College of Pharmacy and Nutrition, University of Saskatchewan, Saskatoon, SK S7N 2Z4, Canada; 10Division of Endocrinology and Metabolism, St. Michael’s Hospital, Toronto, Ontario M5C 2T2, Canada; 11Li Ka Shing Knowledge Institute, St. Michael’s Hospital, Toronto, ON M5C 2T2, Canada

**Keywords:** dietary approaches to stop hypertension, DASH, cardiometabolic health, cardiovascular disease, review, GRADE

## Abstract

Background: The Dietary Approaches to Stop Hypertension (DASH) dietary pattern, which emphasizes fruit, vegetables, fat-free/low-fat dairy, whole grains, nuts and legumes, and limits saturated fat, cholesterol, red and processed meats, sweets, added sugars, salt and sugar-sweetened beverages, is widely recommended by international diabetes and heart association guidelines. Objective: To summarize the available evidence for the update of the European Association of the Study of Diabetes (EASD) guidelines, we conducted an umbrella review of existing systematic reviews and meta-analyses using the Grading of Recommendations Assessment, Development, and Evaluation (GRADE) approach of the relation of the DASH dietary pattern with cardiovascular disease and other cardiometabolic outcomes in prospective cohort studies and its effect on blood pressure and other cardiometabolic risk factors in controlled trials in individuals with and without diabetes. Methods: MEDLINE and EMBASE were searched through 3 January 2019. We included systematic reviews and meta-analyses assessing the relation of the DASH dietary pattern with cardiometabolic disease outcomes in prospective cohort studies and the effect on cardiometabolic risk factors in randomized and non-randomized controlled trials. Two independent reviewers extracted relevant data and assessed the risk of bias of individual studies. The primary outcome was incident cardiovascular disease (CVD) in the prospective cohort studies and systolic blood pressure in the controlled trials. Secondary outcomes included incident coronary heart disease, stroke, and diabetes in prospective cohort studies and other established cardiometabolic risk factors in controlled trials. If the search did not identify an existing systematic review and meta-analysis on a pre-specified outcome, then we conducted our own systematic review and meta-analysis. The evidence was summarized as risk ratios (RR) for disease incidence outcomes and mean differences (MDs) for risk factor outcomes with 95% confidence intervals (95% CIs). The certainty of the evidence was assessed using GRADE. Results: We identified three systematic reviews and meta-analyses of 15 unique prospective cohort studies (*n* = 942,140) and four systematic reviews and meta-analyses of 31 unique controlled trials (*n* = 4,414) across outcomes. We conducted our own systematic review and meta-analysis of 2 controlled trials (*n* = 65) for HbA1c. The DASH dietary pattern was associated with decreased incident cardiovascular disease (RR, 0.80 (0.76–0.85)), coronary heart disease (0.79 (0.71–0.88)), stroke (0.81 (0.72–0.92)), and diabetes (0.82 (0.74–0.92)) in prospective cohort studies and decreased systolic (MD, −5.2 mmHg (95% CI, −7.0 to −3.4)) and diastolic (−2.60 mmHg (−3.50 to −1.70)) blood pressure, Total-C (−0.20 mmol/L (−0.31 to −0.10)), LDL-C (−0.10 mmol/L (−0.20 to −0.01)), HbA1c (−0.53% (−0.62, −0.43)), fasting blood insulin (−0.15 μU/mL (−0.22 to −0.08)), and body weight (−1.42 kg (−2.03 to −0.82)) in controlled trials. There was no effect on HDL-C, triglycerides, fasting blood glucose, HOMA-IR, or CRP. The certainty of the evidence was moderate for SBP and low for CVD incidence and ranged from very low to moderate for the secondary outcomes. Conclusions: Current evidence allows for the conclusion that the DASH dietary pattern is associated with decreased incidence of cardiovascular disease and improves blood pressure with evidence of other cardiometabolic advantages in people with and without diabetes. More research is needed to improve the certainty of the estimates.

## 1. Introduction

Cardiovascular disease (CVD) continues to be a leading cause of mortality in people with and without diabetes globally [1,2,3]. Clinical practice guidelines recommend dietary strategies as the cornerstone of therapy to prevent and manage cardiovascular disease [4,5,6,7,8,9]. The dietary approaches to stop hypertension (DASH) dietary pattern, which emphasizes fruit, vegetables, fat-free/low-fat dairy, whole grains, nuts and legumes, and limits total and saturated fat, cholesterol, red and processed meats, sweets, added sugars, and sugar-sweetened beverages, was originally developed through research sponsored by the US National Institutes of Health (NIH) to treat hypertension without medication and successfully demonstrated a clinically meaningful blood pressure lowering effect [10]. 

In addition to reducing blood pressure, the DASH dietary pattern has since been shown to have a decreasing effect on low-density lipoprotein-cholesterol (LDL-C) among other cardiometabolic risk factors in randomized controlled trials and be associated with reductions in diabetes and cardiovascular mortality in prospective cohort studies [11,12,13]. 

These benefits of the DASH dietary pattern have been recognized by general dietary guidelines from the U.S.-based National Heart, Lung, and Blood Institute (NHLBI) and the United States Department of Agriculture (USDA) [14]. International diabetes [7,15] and cardiovascular [6,16,17] clinical practice guidelines have also recommended the DASH dietary pattern for cardiovascular risk reduction. The European Association for the Study of Diabetes (EASD), however, has not reviewed the evidence or made specific recommendations regarding the DASH dietary pattern in its clinical practice guidelines for nutrition therapy. To update current recommendations, the Diabetes and Nutrition Study Group (DNSG) of the EASD commissioned an umbrella review of existing systematic reviews and meta-analyses using the Grading of Recommendations Assessment, Development, and Evaluation (GRADE) approach to summarize the available evidence of the relation of the DASH dietary pattern with diabetes and cardiovascular outcomes in prospective cohort studies and its effect on blood pressure and other established cardiometabolic risk factors in randomized and non-randomized controlled trials.

## 2. Materials and Methods

### 2.1. Design 

Public health policy and clinical practice guidelines are established with the use of systematic reviews and meta-analyses of controlled trials and prospective cohort studies, which are regarded as the best levels of evidence. We thus identified the most recent systematic reviews and meta-analyses assessing the relationships of the DASH dietary pattern with incident cardiometabolic diseases in prospective cohort studies and on cardiometabolic risk factors in randomized and non-randomized controlled trials in individuals with and without diabetes. The umbrella review was conducted according to the principals of the Cochrane Handbook for Systematic Reviews of Interventions [18] and the GRADE handbook [19] with reporting according to the Preferred Reporting Items for Systematic Reviews and Meta-Analyses (PRISMA) [20]. The study protocol was registered (clinicaltrials.gov identifier, NCT03542370).

### 2.2. Study selection

The databases Medline and Embase were searched from inception through 3 January 2019 using the search terms “dietary approaches to stop hypertension” or “DASH” and “meta-analysis” (Appendix A). If the search of systematic reviews and meta-analyses did not identify an existing systematic review and meta-analysis on any of the pre-specified outcomes below, then we conducted our own systematic review and meta-analysis in which studies were eligible if the intervention was a DASH dietary pattern and the outcomes of interest were reported.

### 2.3. Data extraction

Two independent reviewers extracted relevant data from each included systematic review and meta-analysis and from invidual studies if we had to conduct our own systmatic review and meta-analysis.

### 2.4. Risk of Bias Assessment

The quality of the individual studies contained in each systematic review and meta-analysis was assessed by the two independent reviewers. Study characteristics were extracted and risk of bias assessments were performed using either the New Castle Ottawa score [21] for the prospective cohort studies or the Cochrane Risk of Bias Tool [22] for the controlled trials. 

### 2.5. Outomes

The primary outcome was incident CVD in the prospective cohort studies and SBP in the controlled trials. Secondary outcomes included incident coronary heart disease (CHD), stroke, and diabetes in the prospective cohort studies and diastolic blood pressure (DBP), blood lipids (Total-cholesterol (Total-C), LDL-C, high-density lipoprotein-cholesterol (HDL-C), and triglycerides), glycemic control (HbA1c, fasting blood glucose, fasting blood insulin, homeostasis model assessment of insulin resistance (HOMA-IR)), adiposity (body weight), and inflammation (C-reactive protein) in the controlled trials.

### 2.6. Evidence Synthesis

All of the available evidence for each systematic review and meta-analysis identified was summarized, including pooled risk ratios for reports of prospective cohorts and pooled effect estimates of mean differences (MDs) for reports of controlled trials. Where we conducted our own meta-analysis, the generic inverse variance method with fixed or random effects models were used where appropriate [18]. 

### 2.7. Grading of the Evidence

The certainty of the evidence was assessed using the GRADE tool [19,23,24,25,26,27,28,29,30,31,32,33,34]. This tool allows evidence to be graded as high, moderate, low, or very low quality. Randomized controlled trials start as high-quality evidence and observational studies such as prospective cohort studies start as low-quality evidence. Both can then be downgraded or upgraded on the basis of pre-specified criteria. The criteria used to downgrade evidence include study limitations (weight of studies showing risk of bias as assessed by the Cochrane Risk of Bias Tool [22] or the New Castle Ottawa Scale [21], unless otherwise specified), inconsistency (substantial unexplained inter-study heterogeneity, I^2^ ≥ 50% and *P* < 0.10), indirectness (presence of factors that limit the generalizability of the results), imprecision (the 95% confidence intervals (95% CIs) for MDs and risk estimates are wide or cross a minimally important difference), and publication bias (significant evidence of small-study effects). The criteria used to upgrade the quality of evidence are restricted to prospective cohort studies. These criteria include a large magnitude of association (relative risk, (RR) ≤ 0.5 or ≥2), a dose–response gradient, and attenuation by plausible confounding.

## 3. Results

### 3.1. Search Results

Figure 1 illustrates the literature search and selection process. We identified 125 reports from the search, of which 60 were excluded for title and abstract. Of 17 reports that were reviewed in full, seven reports met eligibility criteria and were included. We identified three systematic reviews and meta-analyses of 15 unique prospective cohort studies (*n* = 942,140) and four systematic reviews and meta-analyses of 31 unique controlled trials (*n* = 4414) across outcomes.

### 3.2. Outcomes

#### 3.2.1. Systematic Reviews and Meta-Analyses of Prospective Cohort Studies

##### Cardiovascular Disease Incidence

One systematic review and meta-analysis assessed the relationship between consumption of the DASH dietary pattern and CVD incidence (including incidence and mortality of CVD, CHD, stroke and sudden cardiac death) [13] (Figure 2, Table 1, and Appendix A). It included 11 prospective cohort comparisons [35,36,37,38,39,40,41,42] (*n* = 783,732; 32,927 events) conducted in various countries, including the United States (seven studies), Sweden (two studies), China (one study), and Italy (one study), with follow-up durations ranging from 7.9 to 24 years [13]. The consumption of a DASH dietary pattern was found to significantly reduce CVD (RR = 0.80 (95% CI: 0.76–0.85), which showed no evidence of inter-study heterogeneity (I^2^ = 30%) [13]. No serious risk of bias was identified (Appendix A).

Appendix A shows the GRADE assessment of the certainty of the evidence for the relationship between the DASH dietary pattern and CVD. The evidence was rated as low for the association of the DASH dietary pattern and CVD incidence. This assessment suggests that the DASH dietary pattern may have a meaningful cardiovascular benefit, but the estimate remains uncertain.

##### Coronary Heart Disease Incidence

One systematic review and meta-analysis assessed the relationship between consumption of the DASH dietary pattern and CHD incidence [43] (Figure 2, Table 1, and Appendix A). It included three prospective cohort comparisons [37,38,39] (*n* = 144,337; 7260 events) all of which were conducted in the United States, with follow-up durations ranging from 14.6 to 24 years [43]. The consumption of a DASH dietary pattern was found to significantly reduce CHD incidence (RR = 0.79 (95% CI: 0.71–0.88)), which showed no evidence of inter-study heterogeneity (I^2^ = 0%) [43]. No serious risk of bias was identified (Appendix A).

Appendix A shows the GRADE assessment of the overall strength of the evidence for the relationship between the DASH dietary pattern and CHD incidence. The evidence was rated as very low for the association of the DASH dietary pattern and CHD incidence, owing to a downgrade for indirectness since the findings are not generalizable given that the three prospective cohort studies were conducted in middle-aged or elderly women. The relationship remains uncertain, with future studies likely to have an important influence on risk estimates.

##### Stroke Incidence

One systematic review and meta-analysis assessed the relationship between consumption of the DASH dietary pattern and stroke incidence [43] (Figure 2, Table 1 and Appendix A). It included three prospective cohort comparisons [35,38,39] (*n* = 150,191; 4,413 events) two of which were conducted in the United States and one study in Italy, with follow-up durations ranging from 7.9–24 years [43]. The consumption of a DASH dietary pattern was found to significantly reduce stroke incidence (RR = 0.81 (95% CI: 0.72–0.92)), which showed no evidence of inter-study heterogeneity (I^2^ = 0%) [43]. No serious risk of bias was identified (Appendix A).

Appendix A shows the GRADE assessment of the certainty of the evidence for the relationship between the DASH dietary pattern and stroke incidence. The evidence was rated as low for the association of the DASH dietary pattern and stroke incidence. This assessment suggests that the DASH dietary pattern may have a meaningful stroke benefit, but the estimate remains uncertain.

##### Diabetes Incidence

One systematic review and meta-analysis assessed the relationship between consumption of the DASH dietary pattern and diabetes incidence [12] (Figure 2, Table 1, and Appendix A). It included five prospective cohort studies [44,45,46,47,48] (*n* = 158,408; 23,612 events) four of which were conducted in the United States and one study in Europe, with follow-up durations ranging from 5–20 years [12]. The consumption of a DASH dietary pattern was found to significantly reduce diabetes incidence (RR = 0.82 (95% CI: 0.74–0.92)), however showed substantial unexplained inter-study heterogeneity (I^2^ = 62%) [12]. No serious risk of bias was identified (Appendix A).

Appendix A shows the GRADE assessment of the certainty of the evidence for the relationship between the DASH dietary pattern and diabetes incidence. The evidence was rated as very low for the association of the DASH dietary pattern and diabetes incidence, owing to a downgrade for inconsistency (I² = 62%; *P* = 0.03), and with <10 studies, no subgroup analyses were performed to attempt to explain heterogeneity. The relationship remains uncertain, with future studies likely to have an important influence on risk estimates.

#### 3.2.2. Systematic Reviews and Meta-analyses of Controlled Trials

##### Blood Pressure

One systematic review and meta-analysis of controlled trials assessed the effect of the DASH dietary pattern on blood pressure outcomes, including SBP and DBP [11] (Figure 3, Table 2, and Appendix A). A total of nineteen controlled trials [10,49,50,51,52,53,54,55,56,57,58,59,60,61,62,63] were included, involving 1,918 middle-aged participants with and without hypertension. The DASH dietary pattern was found to significantly lower SBP (MD = −5.20 mmHg (95% CI: −7.00 to −3.40 mmHg)) and DBP (MD = −2.60 mmHg (95% CI: −3.50 to −1.70 mmHg)). There was substantial unexplained inter-study heterogeneity across both outcomes (I^2^ = 76% and 49%, respectively). No serious risk of bias was identified (Appendix A).

Appendix A shows the GRADE assessments for the certainty of the evidence for the effect of a DASH dietary pattern on blood pressure. The evidence for SBP was rated as moderate, owing to a downgrade for inconsistency (I² = 76%; *P* < 0.001). The evidence for DBP was rated as low, owing to downgrades for inconsistency (I² = 49%; *P* = 0.009) and imprecision in the pooled effect estimate. This assessment suggests that the DASH dietary pattern may result in clinically meaningful reductions in blood pressure. The effect of the DASH dietary pattern on DBP, however, remains uncertain, with future randomized controlled trials likely to have an important influence on risk estimates.

##### Blood Lipids

One systematic review and meta-analysis of controlled trials assessed the effect of the DASH dietary pattern on blood lipid outcomes, including Total-C, LDL-C, HDL-C and triglycerides [11] (Figure 3, Table 2, and Appendix A). A total of thirteen controlled trials [50,52,53,54,55,57,58,59,63,67,68] were included in the analysis of Total-C and LDL-C, involving 1673 middle-aged participants, fifteen trials [50,52,53,54,55,56,57,58,59,63,67,68] in the analysis of HDL-C, involving 1749 participants, and 14 trials [50,52,54,55,56,57,58,59,63,67,68] in the analysis of triglycerides, involving 1654 participants. The DASH dietary pattern was found to lower Total-C (MD = −0.20 mmol/L (95% CI: −0.31 to −0.10 mmol/L)) and LDL-C (MD = −0.10 mmol/L (95% CI: −0.20 to −0.01 mmol/L)) with no significant effects on HDL-C or triglycerides. There was substantial unexplained inter-study heterogeneity for Total-C and HDL-C (I^2^ = 52% and 76%, respectively), some evidence of inter-study heterogeneity for LDL-C (I^2^ = 37%) and no evidence of inter-study heterogeneity for triglycerides (I^2^ = 0%). No serious risk of bias was identified (Appendix A).

Appendix A shows the GRADE assessments for the certainty of the evidence for the effect of a DASH dietary pattern on blood lipids. The evidence for LDL-C was rated as moderate, owing to a downgrade for imprecision in the pooled effect estimate. The evidence for Total-C, HDL-C, and triglycerides were rated as low, where Total-C and HDL-C were downgraded for inconsistency (I² = 52% and 76%, respectively) and imprecision in the pooled effect estimates and triglycerides was downgraded for imprecision in the pooled effect estimate and for evidence of potential publication bias. This assessment suggests that the DASH dietary pattern may result in reductions in Total-C and LDL-C, established therapeutic lipid targets for cardiovascular risk reduction. However, sources of uncertainty remain. Thus, there is a need for further large, high quality, randomized controlled trials to clarify the lipid-lowering benefits of the DASH dietary pattern.

##### Glycemic Control

There were no systematic reviews and meta-analyses identified of trials assessing the effect of the DASH dietary pattern on HbA1c. Therefore, we conducted a systematic review and meta-analysis for this outcome. The search (Appendix A) identified 2 controlled trials [57,63] (Appendix A) which were eligible for inclusion, involving 65 middle-aged participants. The DASH dietary pattern was found to lower HbA1c (MD = −0.53% (95% CI: −0.62 to −0.43%) with significant evidence of inter-study heterogeneity (I^2^ = 99%) (Figure 3, Table 2, and Appendix A).

One systematic review and meta-analysis of controlled trials assessed the effect of the DASH dietary pattern on blood glucose [11], insulin, and HOMA-IR [64] (Figure 3, Table 2, and Appendix A). A total of ten controlled trials [50,52,56,57,58,59,63] were included in the analysis of blood glucose, involving 826 middle-aged participants, 11 trials [52,58,59,69,70,71,72] in the analysis of insulin, involving 760 participants, and eight trials [52,58,70,71] in the analysis of HOMA-IR, involving 603 participants. The DASH dietary pattern was found to lower insulin (MD = −0.15 μU/mL (95% CI: −0.22 to −0.08 μU/mL)) with no significant effects on blood glucose or HOMA-IR. There was no evidence of inter-study heterogeneity for insulin (I^2^ = 0%) or for HOMA-IR (I^2^ = 16%). There was substantial unexplained inter-study heterogeneity for blood glucose (I^2^ = 59%) No serious risk of bias was identified (Appendix A).

Appendix A shows the GRADE assessments for the certainty of the evidence for the effect of a DASH dietary pattern on glycemic control. The evidence for HbA1c was rated as low, owing to downgrades for inconsistency (I² = 99%) and serious indirectness due to <5 studies available for inclusion and lack of generalizability since one study included those with type 2 diabetes and the other women with gestational diabetes. The evidence for blood glucose was also rated as low, owing to downgrades for inconsistency (I^2^ = 59%) and imprecision in the pooled effect estimate. The evidence for fasting insulin and HOMA-IR were rated as moderate, owing to downgrades for imprecision in the pooled effect estimates. This assessment suggests that the DASH dietary pattern may result in reductions in HbA1c. The effect estimates, however, remain uncertain for most glycemic outcomes, calling for more large, high-quality, randomized trials to clarify the glycemic benefits.

##### Body Weight

One systematic review and meta-analysis of controlled trials assessed the effect of the DASH dietary pattern on body weight [65] (Figure 3, Table 2, and Appendix A). A total of 11 controlled trials [53,55,56,59,73,74,75,76,77,78] were included in the analysis of body weight, involving 1211 middle-aged participants. The DASH dietary pattern was found to lower body weight (MD = −1.42 kg (95% CI: −2.03 to −0.82 kg)). There was substantial unexplained inter-study heterogeneity for body weight (I^2^ = 71%). No serious risk of bias was identified (Appendix A).

Appendix A shows the GRADE assessments for the certainty of the evidence for the effect of a DASH dietary pattern on body weight. The evidence for body weight was rated as moderate, owing to a downgrade for inconsistency (I² = 71%). This assessment suggests that the DASH dietary pattern may result in meaningful reductions in body weight. The relationship, however, remains uncertain, with future randomized controlled trials likely to have an important influence on risk estimates.

##### Inflammation

One systematic review and meta-analysis of controlled trials assessed the effect of the DASH dietary pattern on inflammation [66] (Figure 3, Table 2, and Appendix A). A total of six controlled trials [57,77,78,79,80,81] were included in the analysis of the inflammatory marker C-reactive protein (CRP), involving 451 middle-aged participants. The DASH dietary pattern was found not to have an effect on CRP. There was substantial unexplained inter-study heterogeneity for CRP (I^2^ = 97%). However, subgroup analyses performed by Soltani et al. [66] based on comparator revealed a significant effect compared to unhealthy or usual diets (4 studies) ((MD = −9.62 nmol/L (95% CI: −15.62 to −3.62 nmol/L), I^2^ = 67.7%), as well as based on follow-up where there was an effect in trials with ≥8 weeks follow-up ((MD = −7.05 nmol/L (95% CI: −12.95 to −1.05 nmol/L), I^2^ = 92.7%). No serious risk of bias was identified (Appendix A).

Appendix A shows the GRADE assessments for the certainty of the evidence for the effect of a DASH dietary pattern on CRP. The evidence for CRP was rated as low, owing to downgrades for inconsistency (I^2^ = 97%) and imprecision in the pooled effect estimate. This assessment suggests uncertainty in whether the DASH dietary pattern has an effect on inflammation. Future randomized controlled trials are likely to have an important influence on risk estimates.

## 4. Discussion

The present umbrella review of the DASH dietary pattern and cardiometabolic outcomes identified three systematic reviews and meta-analyses of prospective cohort studies involving 15 unique cohort comparisons in 942,140 participants and 32,927 CVD events, 7260 CHD events, 4413 stroke events, 23,612 diabetes events, and four systematic reviews and meta-analyses supplemented by one updated systematic review and meta-analysis of randomized and non-randomized controlled trials involving 33 trial comparisons in 4479 participants on intermediate cardiometabolic risk factors. The DASH dietary pattern was associated with a reduction in the primary outcome of the prospective cohort studies, CVD incidence (20%), as well as reductions in the secondary outcomes: CHD (21%), stroke (19%), and diabetes (18%). These changes were supported by a clinically meaningful reduction in the primary outcome of the controlled trials, blood pressure (−5.2 mmHg for SBP), as well as reductions in the secondary outcomes: DBP (−2.6 mmHg), lipids (−0.1 mmol/L for LDL-C and −0.2 mmol/L for Total-C), body weight (1.42 kg), and HbA1c (−0.53%).

### 4.1. Findings in the Context of the Literature

The observed cardiovascular benefits may be attributable to a combination of the foods encouraged as part of the DASH dietary pattern. High consumption of fruits and vegetables as part of the DASH dietary pattern have been shown in systematic reviews and meta-analyses of prospective cohort studies, when consumed either together or alone, to be inversely associated with cardiovascular incidence and mortality [82]. Systematic reviews and meta-analyses of prospective cohort studies have also demonstrated that whole grain intake is associated with a 20–21% reduction in CVD incidence [83,84] and 14% reduction in stroke incidence [84], dietary pulse intake with a 9% reduction in CVD incidence [85], and nut intake with a 21% reduction per 28 g/day [86], while processed and red meats are associated with a 15–18% increase in incidence of CVD mortality comparing highest to lowest levels of intake [87,88]. A key nutrient richly found in many of the foods emphasized which may contribute a biological effect includes dietary fibre, which has been shown in systematic reviews and meta-analyses of prospective cohort studies to reduce CVD by 9% per 7g/d intake [89] and reduced risks of CHD and stroke incidence of 24% and 22%, respectively, when comparing highest to lowest intake groups [84], in addition to significant reductions in body weight, Total-C, LDL-C, and SBP in controlled clinical trials [84]. CVD benefit may be the result of the biological effects of other key nutrients richly found in foods emphasized in the DASH dietary pattern, such as magnesium, potassium, and phytochemicals, including flavonoids, which have been demonstrated to have anti-inflammatory and anti-antioxidant activity and result in reductions in angiogenesis [56,90,91]. The blood pressure lowering effect of the DASH diet may play a major role, since high blood pressure is ranked as the strongest risk factor attributable to chronic disease [10,92]. Furthermore, the observed 5.20-mmHg lowering in SBP by the DASH dietary pattern is clinically relevant based on evidence from prospective studies showing that a 2-mmHg reduction in SBP is associated with lower mortality from stroke (10%) and CHD or other vascular causes (7%) in middle-aged men and women [93]. An additional contribution may be via a reduction in established therapeutic lipid targets for cardiovascular risk [11], such as the observed 0.10mmol/L lowering in LDL-C, which would translate to about a 2% reduction in major cardiovascular events based on the Cholesterol Treatment Trialists' (CTT) Collaboration [94,95,96]. The observed overall 1.42 kg reduction in body weight may be considered clinically relevant [97] and may also play a role since obesity is a key risk factor for CVD [98]. Furthermore, a recent prospective cohort not captured in the systematic review and meta-analysis on the DASH dietary pattern and composite CVD outcomes, but which also supports the findings, showed an 11% reduced risk of all-cause mortality when comparing those participants who had the greatest improvement in DASH diet quality score over a 12-year follow-up period compared to those with a relatively stable diet quality [99]. Importantly, they also found a 9% reduced risk among those who maintained a high-quality DASH diet score over the 12-year period compared to those with consistently low diet scores over time [99]. These results highlight the CVD benefit of not only adopting or increasing adherence to a DASH dietary pattern, but also to maintaining a high compliance to it.

The observed reductions in diabetes incidence and improvements in glycemic control again may be attributable to the high consumption of fruit and vegetables as well as the low-fat dairy component of the DASH dietary pattern. Systematic reviews and meta-analyses have shown that fruit and vegetables alone or together [100] and low-fat milk and yogurt [101] are associated with reduced diabetes incidence. Again, dietary fibre coming from many of the foods may play a role since a recent systematic review and meta-analysis of prospective cohort studies found a 16% reduction in incidence of type 2 diabetes when comparing the highest to lowest fibre intakes^84^. The blood pressure lowering effects of the DASH diet may contribute to this effect since hypertension is associated with type 2 diabetes [102], where prospective cohort studies have demonstrated that a 1-mmHg increase in SBP is associated with a 1%–4% increase in type 2 diabetes risk [103,104]. The observed reduction in body weight may also contribute since body weight is strongly associated with diabetes risk [98]. The increase in legumes, nuts, fruit, whole grains and dietary fibre intake, especially from whole grain viscous fibres, as part of a DASH diet may also be the mechanism by which there is a reduction in HbA1c, since these have been demonstrated to improve glycemic control in systematic reviews and meta-analyses of randomized controlled trials [105,106,107,108,109] and have been associated with reduced diabetes incidence in systematic reviews and meta-analyses of prospective cohort studies [110,111]. 

The DASH diet was originally developed to contain foods which increase magnesium, potassium, and calcium based on established links to lower blood pressure and successfully demonstrated a clinically meaningful blood pressure lowering effect [10]. Interactions between these nutrients can also have blood pressure lowering effects, such as the sodium-to-potassium ratio or the interaction between potassium and calcium and the ability to increase sodium excretion by the kidneys [112]. Potassium and calcium have previously been demonstrated to interact with the renin-angiotensin system by affecting plasma renin activity [113,114,115] and potassium can assist with sodium balances and has also been demonstrated to potentially lower blood pressure through endothelium-dependent vascular effects [115,116,117]. Additionally, it has been suggested that a reduction in blood pressure may result from an increased intake of nitrate-rich foods, including fruits and vegetables, especially leafy vegetables, via the role of inorganic nitrate in the non-enzymatic generation of nitric oxide [118]. Furthermore, many of the foods encouraged on the DASH diet have demonstrated blood pressure reductions in systematic reviews and meta-analyses of controlled trials in those with and without diabetes, including legumes [119], fruit [120], and whole grains, particularly those rich in viscous fibres [106,121].

The observed reductions in lipids, including the primary lipid target for therapy, LDL-C, may be attributable to the high consumption of fruit, nuts, legumes, and whole grains (especially from oats and barley), increases in dietary fibre, and reductions in saturated fat intake as part of the DASH dietary pattern. Systematic reviews and meta-analyses of randomized controlled trials have shown that each of these alone [120,122,123,124,125] or combined as part of cholesterol-lowering dietary patterns such as the Portfolio dietary pattern [126] lowers Total-C and LDL-C.

The effect of the DASH dietary pattern on body weight may result from increased fruit and vegetable consumption, as was found in a systematic review and meta-analysis of randomized controlled trials [127]. Greater dietary fibre intake may contribute to weight loss since high-fibre foods require longer chewing time and promote gastric distention, triggering signals of fullness and slowed digestion, and delayed absorption of nutrients could delay hunger and subsequent energy intake [128]. Increasing dietary pulse intake has also been demonstrated in systematic reviews and meta-analyses of controlled trials to reduce body weight [129] and increase satiety [130]. Reduced sodium intake may also play a role in lowering body weight since high sodium intake is associated with obesity in the general population, possibly because of the associated increase in thirst and appetite [131].

Although a significant effect on CRP was not observed in the primary pooled analyses, subgroup analyses demonstrated that when the DASH diet was compared to an unhealthy or usual diet as opposed to a healthy diet (e.g. Portfolio diet) or when the follow up duration was ≥8 weeks, the DASH diet resulted in a significantly lower CRP. If this is a true effect, then the reduction in inflammation may also play a key role in the observed reduction in incidence of cardiometabolic diseases. The effect may be mediated by increased dietary fibre intake [132] possibly because of the delay in glucose absorption and alteration to gut microflora which may suppress inflammatory cytokines production, stimulate the production of short-chain fatty acids and lead to lower circulating free fatty acid concentrations and, thus, subsequent inflammation [133,134,135]. The effect may also be the result of increased fruit and vegetable intake due to possible anti-inflammatory effects [136,137] and increased magnesium intake [138,139]. The high content of vitamin C, calcium and magnesium coming from fruits and vegetables in the DASH dietary pattern may reduce inflammation through reductions in oxidative stress via decreased NADPH oxidase activity [140] and by restoring the activity of anti-oxidative enzymes [141]. 

### 4.2. Strengths and limitations

The strengths of the current umbrella review include that the included systematic reviews and meta-analyses were all conducted recently with the census dates of each ranging from January 2012 to November 27, 2018 and an assessment of the overall certainty of the evidence was performed using the GRADE approach. The limitations include: indirectness for CHD incidence due to the inclusion of cohorts limited to middle-aged and elderly women and for HbA1c due to the inclusion of 2 trials in either type 2 diabetes or gestational diabetes; unexplained inconsistency for diabetes incidence (I² = 62%; *P* = 0.03) and for 8/12 risk factors; imprecision for 9/12 risk factors; and publication bias for triglycerides. Although we did not downgrade the evidence for indirectness, concern may be raised that many of the included trials and cohorts were conducted in people without diabetes. We did not feel that there was any biological reason to believe that the DASH dietary pattern would behave differently in people with diabetes, as many components of the DASH dietary pattern have been shown individually to lower blood pressure and other established CVD risk factors in systematic reviews and meta-analyses of randomized controlled trials inclusive of people with diabetes without any evidence of a subgroup effect by diabetes status [85,108,109,121,122,129,142,143].We also felt that this concern was mitigated by the evidence of similar or greater improvements in all of the measured outcomes (SBP, DBP, LDL-C, Total-C, HDL-C, TG, HbA1c, body weight, CRP) in those trials conducted exclusively in diabetes (that is, the effect estimates for these outcomes in the individual trials were contained within or exceeded the 95% confidence intervals of the overall pooled estimates). 

Weighing the strengths and limitations, the certainty of the evidence based on the GRADE approach was rated as very low to low for associations with cardiometabolic disease incidence and low to moderate for effects on cardiometabolic risk factors.

### 4.3. Implications

Clinical practice guidelines recommend dietary strategies as the cornerstone of the prevention and management of CVD [4,5,6,7,8,9]. Our pooled analyses demonstrate that the DASH dietary pattern is associated with a 20% reduced CVD incidence and has blood pressure benefits which may translate to about a 20% reduction in risk of CVD, along with meaningful benefits in other established CVD risk factors in those with and without diabetes. In the systematic review and meta-analysis on the DASH dietary pattern and blood pressure, trials included participants who were hypertensive but not on medication [10,49,53,56,58,59,60], as well as trials where participants were taking blood pressure medications [51,54], all of which found significant blood pressure lowering effects. Thus, the DASH dietary pattern may play a role both as a first line therapy as well as an add-on therapy.

There may be an important opportunity for people with and without diabetes to realize the CVD benefits of a DASH dietary pattern. The DASH dietary pattern emphasizes fruit, vegetables, low-fat dairy, whole grains, nuts and legumes, and limits total and saturated fat. Specifically, the DASH diet includes 4–5 servings of fruits and vegetables per day, 2–3 servings of low fat dairy, 6–8 servings of whole grains and limits meat, poultry and fish to less than six servings per day. It also recommends 4–5 servings per week of nuts, seeds, dry beans, and peas and choosing foods with low saturated fat, high potassium and fibre, and low sodium. Dietary intake patterns in Europe and other Western countries do not currently meet these targets. The European Health Interview Survey (EHIS) (Eurostat 2016) reports that, on average, more than a third of the EU adults do not consume any fruits and vegetables on a daily basis and only 14.1% consume five portions per day [144]; about one third of those aged 35 and older had average intakes of saturated fat ≥15 E% [145]; although a high total consumption of dairy products was reported in the Dutch, Swedish and Danish, and most of Spain and the UK, a somewhat low consumption was reported Greece and in some of Italy (Ragusa and Turin) [146]. According to the most recent survey in the United States, only one in 10 adults and youths eat the recommended amount of fruits or vegetables [147], only 1.9 servings/day of dairy products are consumed on average [148], only ~38.2% of adults consume nuts on a given day [149] and most exceed the recommendations for added sugars, saturated fats, and sodium [14]. These data suggest that in these populations that there is an opportunity for people with and without diabetes to increase these foods to achieve a DASH dietary pattern and realize the cardiometabolic benefits.

## 5. Conclusions

In conclusion, this synthesis of systematic reviews and meta-analyses demonstrates that the DASH dietary pattern as a well-accepted blood pressure-lowering diet has associated CVD benefit supported by reductions in blood pressure, HbA1c, LDL-C and other established CVD risk factors in people with and without diabetes. The certainty of the evidence based on the GRADE approach was very low to low for associations with cardiometabolic disease incidence and low to moderate for effects on cardiometabolic risk factors. More research is needed to improve the estimates and confirm that these benefits do translate into reductions in clinical outcomes of clinical practice and public health importance. In this regard, there remains a need for large randomized trials of the effect of the DASH dietary pattern on clinical CVD outcomes in those with and without diabetes. The available evidence does support a potential opportunity for those with and without diabetes to adopt the DASH dietary pattern to improve cardiometabolic health.

## Figures and Tables

**Figure 1 nutrients-11-00338-f001:**
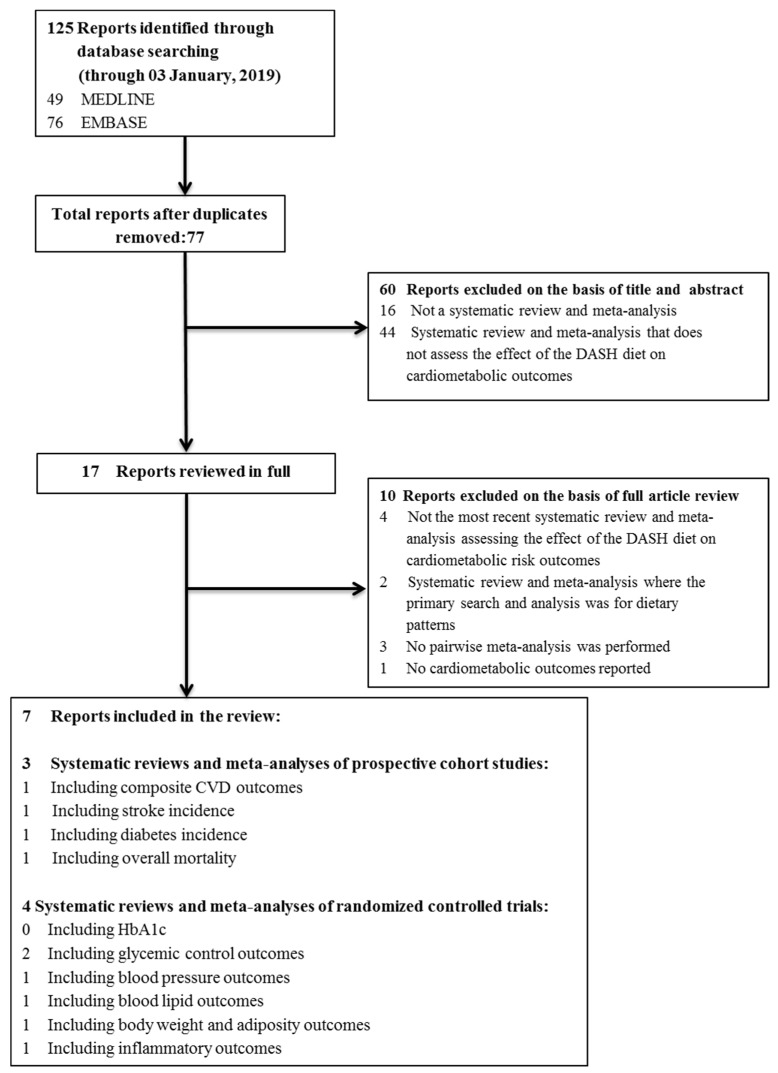
Literature search.

**Figure 2 nutrients-11-00338-f002:**
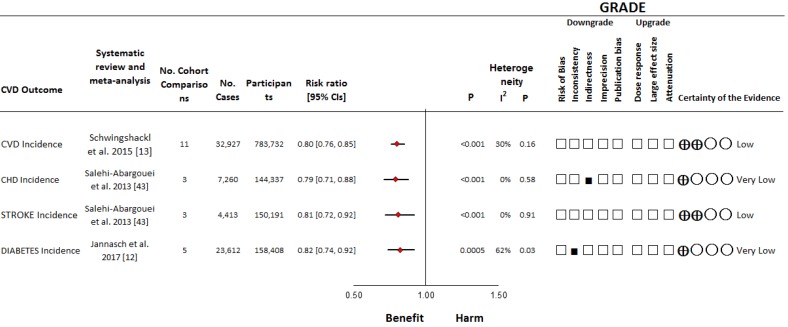
Summary plot of the association between the DASH dietary pattern on risk of various chronic diseases in prospective cohort studies. The pooled risk estimate is represented by the diamond. P-values were determined using random effects modelling in each systematic review and meta-analysis. Between-study heterogeneity was assessed by the Cochran Q statistic, where *P* < 0.10 is considered statistically significant, and quantified by the I^2^ statistic, where I^2^ ≥ 50% is considered evidence of substantial heterogeneity [29]. The Grading of Recommendations, Assessment, Development and Evaluation (GRADE) of prospective cohort studies are rated as “Low” certainty of evidence and can be downgraded by five domains and upgraded by three domains. The filled black squares indicate downgrade and/or upgrades for each outcome. CHD = coronary heart disease; CI = confidence interval; CVD = cardiovascular disease; GRADE = Grading of Recommendations, Assessment, Development and Evaluation; NA = not applicable.

**Figure 3 nutrients-11-00338-f003:**
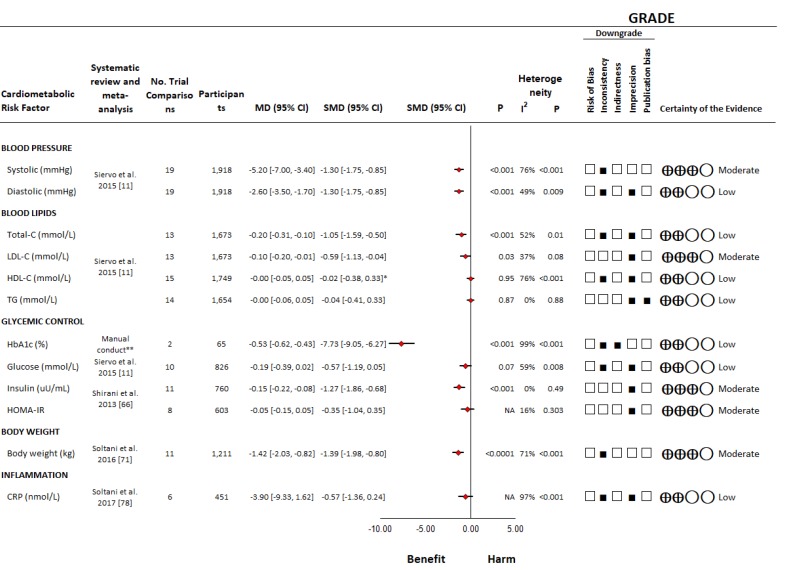
Summary plot of the effect of the DASH dietary pattern on cardiometabolic risk factors in controlled trials. Data are expressed as weighted mean differences with 95% CIs using random effects models in each systematic review and meta-analysis with the exception of HbA1c in which a fixed effects model was used due to the inclusion of <5 trials. To allow the pooled effect estimates for each end point to be displayed on the same axis, mean differences were transformed to standardized mean differences (SMDs). Pseudo-95% CIs for each transformed SMD were derived directly from the original mean difference and 95% CI. Between-study heterogeneity was assessed by the Cochran Q statistic, where *P* < 0.10 is considered statistically significant, and quantified by the I^2^ statistic, where I^2^ ≥ 50% is considered evidence of substantial heterogeneity [29]. The Grading of Recommendations, Assessment, Development and Evaluation (GRADE) of randomized controlled trials are rated as "High" certainty of evidence and can be downgraded by five domains. The filled black squares indicate downgrades for each outcome. *Due to the difference in directionality of HDL-C compared to the other outcomes with regards to signal for benefit or harm, the sign for the SMD was changed. ** Since no published systematic review and meta-analysis was retrieved from the search, we manually conducted a systematic review and meta-analysis on the DASH dietary pattern and HbA1c (Appendix A and Appendix A). To convert Total-C, LDL-C, and HDL-C to mg/dL, multiply by 38.67; to convert TG to mg/dL, multiply by 88.57; to convert blood glucose to mg/dL, multiply by 18.02; to convert CRP to mg/L, multiply by 0.105. CI = confidence interval; CRP = C-reactive protein; GRADE = Grading of Recommendations, Assessment, Development and Evaluation; HbA1c = hemoglobin A1c; HDL-C = high-density lipoprotein-cholesterol; HOMA-IR = Homeostatic Model Assessment of Insulin Resistance; LDL-C = low-density lipoprotein-cholesterol; MD = mean difference; NA = not available; SMD=standardized mean difference; TG = triglycerides; Total-C = total-cholesterol

**Table 1 nutrients-11-00338-t001:** Summary of characteristics of included studies in most recent systematic reviews and meta-analyses of prospective cohort studies assessing the effect of the DASH dietary pattern on chronic disease.

Chronic Disease	Systematic Review and Meta-Analysis	Total no. of obs Studies	Total no. of Participants	Median no. Participants (Range)	Countries	Outcomes Assessed	Total no. of Incident Cases (Range)	Median Age, yr (Range)	Median Duration of Study, yr (Range)	Dietary Intake Assessments (at Baseline)	DASH Exposure Assessments	Method of Outcome Assessment	Funding Source ‡	Risk of Bias Assessment **
CVD	Schwingshackl et al., 2015 [13]	11	783,732	44,544 (2061–242,321)	7 USA: 2 Sweden: 1 Italy: 1 China	3 CHD incidence: 3 CVD mortality: 3 stroke incidence: 1 CVD incidence: 1 sudden cardiac death	32,927 (123–15,497)	60.5 (20–83)	14.6 (7.9-24)	FFQ	7 quintiles: 2 quartiles: 2 tertiles of DASH score	Record linkage	8 Agency: 2 not reported: 1 Agency, Industry	10 H: 1 Lo
CHD	Salehi-Abargouei et al., 2013 [43]	3	144, 337	348,827 (20,993–88,517)	3 USA	1 CHD morbidity and mortality: 1 CHD mortality: 1 fatal and nonfatal CHD	7260 (430–6210)	52 (30–69)	16 (14.6-24)	FFQ	3 quintiles	Record linkage	3 Agency	2 H: 1 Lo
Stroke	Salehi-Abargouei et al., 2013 [43]	3	150,191	40,681 (20,993–88,517)	2 USA: 1 Italy	1 stroke incidence: 1 stroke mortality: 1 fatal and nonfatal stroke	4,413 (178–3999)	52 (30–74)	16 (7.9-24)	FFQ	2 quintiles: 1 tertiles	Record linkage	2 Agency: 1 Agency, Industry	2 H: 1 Lo
Diabetes	Jannasch et al., 2017 [12]	5	158,408	21, 616 (822–89,195)	4 US: 1 Europe (8 countries)	diabetes incidence	23,612 (129–11, 217)	58 (25–84)	11.5 (5-20)	FFQ	4 quintiles: 1 tertiles	3 self-reported + record linkage: 1 independent assessment: 1 OGTT or DM med use	4 Agency: 1 Agency, Industry	4 H: 1 Lo

^‡^ Agency funding is that from government, university or not-for-profit health agency sources. ** Newcastle Ottawa quality assessment Scale was used to assess risk of bias across the following domains: selection (four points), comparability (two points), and outcome (three points). A total score of six or greater was considered high-quality and a total score of five or smaller was considered low-quality. CHD, coronary heart disease; CVD, cardiovascular disease; obs, prospective cohort; DASH, dietary approaches to stop hypertension; DM, diabetes; H, High; Lo, Low; med, medication; obs, observational; OGTT, oral glucose tolerance test; yr, year.

**Table 2 nutrients-11-00338-t002:** Summary of characteristics of included trials in the most recent systematic reviews and meta-analyses of controlled trials assessing the effect of the DASH dietary pattern on cardiometabolic risk factors.

Cardiometabolic Risk Factor	Systematic Review and Meta-Analysis	Total no. of Trials	Total *n*	Median Sample Size (Range)	Metabolic Phenotypes: no. of Trials	Median Age, yr (Range^a^)	Median Follow-up, wks (range)	Trial Design: no. of Trials	Countries: no. of Trials	Randomized: no. of Trials	Intervention: no. of Trials	Comparator: no. of Trials	Feeding/Compliance: no. of Trials	Rob: no. of Trials ^b^
Blood Pressure (SBP + DBP)	Siervo et al., 2015 [11]	19	1, 918	37 (12–537)	9 HTN: 3 PreHTN + HTN: 2 MetS: 1 GDM: 1 Norm: 1 OH, lean: 1 OB: 1 T2DM	44.3 (30.1–59.2)	8 (2–24)	6 CO: 13 P	12 USA: 4 Iran: 3 Australia	16 Y: 3 N	15 DASH alone: 2 weight loss DASH: 1 weight loss DASH + exercise: 1 behavioural intervention plus DASH	10 usual diet: 2 low fat diet: 2 low antioxidant diet: 2 weight loss diet:1 behavioural intervention alone: 1 exercise alone: 1 usual GDM practice	14 dietary advice: 4 metabolic: 1 dietary advice plus some supplemented foods	11 U: 8 Lo
Total-C + LDL-C	Siervo et al., 2015 [11]	13	1, 673	54 (12–537)	6 HTN: 2 PreHTN + HTN: 1 GDM: 1 Norm: 1 OH, Lean: 1 OB: 1 T2DM	48.3 (30.1–59.2)	4 (3–24)	6 CO: 7 P	8 USA: 3 Australia: 2 Iran	10 Y: 3 N	12 DASH alone: 1 behavioural intervention plus DASH	7 usual diet: 2 low fat diet: 2 low antioxidant diet:1 behavioural intervention alone: 1 usual GDM practice	10 dietary advice: 2 metabolic: 1 dietary advice plus some supplemented foods	9 U: 4 Lo
HDL-C	Siervo et al., 2015 [11]	15	1, 749	54 (12–537)	6 HTN: 2 PreHTN + HTN: 2 MetS: 1 GDM: 1 Norm: 1 OH, lean: 1 OB: 1 T2DM	44.0 (30.1–59.2)	8 (3–24)	6 CO: 9 P	8 USA: 4 Iran: 3 Australia	12 Y: 3 N	12 DASH alone: 2 weight loss DASH: 1 behavioural intervention plus DASH	7 usual diet: 2 low fat diet: 2 low antioxidant diet: 2 weight loss diet:1 behavioural intervention alone: 1 usual GDM practice	12 dietary advice: 2 metabolic: 1 dietary advice plus some supplemented foods	11 U: 4 Lo
Triglycerides	Siervo et al., 2015 [11]	14	1, 654	44 (12–537)	5 HTN: 2 PreHTN + HTN: 2 MetS: 1 GDM: 1 Norm: 1 OH, Lean: 1 OB: 1 T2DM	42.6 (30.1–55.6)	6 (3–24)	6 CO: 8 P	8 USA: 4 Iran: 2 Australia	11 Y: 3 N	11 DASH alone: 2 weight loss DASH: 1 behavioural intervention plus DASH	7 usual diet: 1 low fat diet: 2 low antioxidant diet: 2 weight loss diet:1 behavioural intervention alone: 1 usual GDM practice	12 dietary advice: 2 metabolic	10 U: 4 Lo
HbA1c	Manual conduct^c^	2	65	33 (31–34)	1 GDM: 1 T2DM	42.6 (30.1–55.0)	6 (4–8)	1 CO: 1P	2 Iran	2 Y	2 DASH alone	2 usual diet	2 dietary advice	1 U: 1 Lo
Blood glucose	Siervo et al., 2015 [11]	10	826	27 (12–537)	1 HTN: 2 PreHTN + HTN: 2 MetS: 1 GDM: 1 Norm: 1 OH, Lean: 1 OB: 1 T2DM	40.8 (30.1–55.0)	6 (3–24)	5 CO: 5 P	6 USA: 4 Iran	8 Y: 2 N	7 DASH alone: 2 weight loss DASH: 1 behavioural intervention plus DASH	4 usual diet: 2 low antioxidant diet: 2 weight loss diet: 1 behavioural intervention alone: 1 usual GDM practice	10 dietary advice	7 U: 3 Lo
Fasting insulin	Shirani et al., 2013 [64]	11	760	15 (9–266)	2 HTN: 4 PreHTN + HTN: 1 Norm: 1 OH: 1 OH, Lean: 2 OW/OB	44.1 (34.3–51.8)	4 (3–24)	6 CO: 5 P	10 USA: 1 UK	6 Y: 5 N	8 DASH alone: 3 behavioural intervention plus DASH	6 usual diet: 2 low antioxidant diet: 3 advice only	10 dietary advice: 1 dietary advice plus some supplemented foods	7 U: 4 Lo
HOMA-IR	Shirani et al., 2013 [64]	8	603	14 (9–266)	1 HTN: 3 PreHTN + HTN: 1 Norm: 1 OH: 1 OH, Lean: 1 OB	39.7 (34.3–49.8)	3.5 (3–24)	6 CO: 2 P	8 USA	4 Y: 4 N	6 DASH alone: 2 behavioural intervention plus DASH	4 usual diet: 2 low antioxidant diet: 2 advice only	8 dietary advice	6 U: 2 Lo
Body weight	Soltani et al., 2016 [65]	11	1,211	54 (22–476)	5 HTN: 2 MetS: 1 PreHTN + HTN: 1 HF patients: 1 OW/OB, NAFLD: 1 OW/OB, PCOS	48.5 (30.1–62.0)	16 (8–52)	0 CO: 11 P	4 USA: 4 Iran: 2 Australia: 1 Brazil	11 Y: 0 N	5 DASH alone: 4 weight loss DASH: 1 behavioural intervention Plus DASH: 1 DASH + LGI	4 weight loss: 2 low fat: 2 usual diet: 1 behavioural intervention: 1 general HF recommendations: 1 standard low sodium HTN advice	10 dietary advice: 1 dietary advice plus some supplemented foods	6 U: 5 Lo
CRP	Soltani et al., 2017 [66]	6	451	42 (31–241)	2 Hyperlipidemic: 1 Lean Norm + OB HTN: 1 OW/OB, NAFLD: 1 OW/OB, PCOS: 1 T2DM	45.7 (30.1–55.0)	8 (3–24)	3 CO: 3 P	3 Iran: 2 USA: 1 Canada	16 Y: 0 N	3 DASH alone: 2 weight loss DASH: 1 lacto-ovo vegetarian DASH	2 weight loss: 1 usual diet: 1 usual plus fibre, potassium, magnesium: 1 healthy American: 1 Portfolio diet (plant-based with soy protein, viscous fibres and nuts)	15 dietary advice: 1 metabolic	3 U: 3 Lo

A range represents the range of the mean age in the trials. b For ROB, an assessment was performed using the Cochrane Risk of Bias tool, including the evaluation of individual domains of risk of bias (sequence generation, allocation concealment, blinding of participants/personnel and outcome assessors, incomplete outcome data, selective outcome reporting). Each of the five domains was evaluated as either low, high or unclear ROB and the overall ROB category was determined based on the most selected category. c Since no published systematic review and meta-analysis was retrieved from the search, we manually conducted a systematic review and meta-analysis on the DASH dietary pattern and HbA1c (Appendix A and Appendix A). BMI, body mass index; BP, blood pressure; CO, crossover; CRP, C-reactive protein; DASH, dietary approaches to stop hypertension; DBP, diastolic blood pressure; DM, diabetes; F, female; F/U, follow-up; GDM, gestational diabetes; HDL-C, high-density lipoprotein-cholesterol; HF, heart failure; HTN, hypertensive; HOMA-IR, Homeostatic Model Assessment of Insulin Resistance; HTN, hypertension; L, lean; Lo, Low; LDL-C, low-density lipoprotein-cholesterol; LGI, low glycemic index; M, male; meds, medication; MetS, metabolic syndrome; N, no; Norm, normotensive; NAFLD, non-alcoholic fatty liver disease; OB, obese; OH, overall healthy; OW, overweight; P, parallel; PCOS, polycystic ovarian syndrome; PreHTN, prehypertensive; ROB, Risk of Bias; SBP, systolic blood pressure; SD, standard deviation; T2DM, type 2 diabetes; Total-C, total-cholesterol; U, unclear; W, women; wks, weeks; Y, yes; yr; year.

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
