# Peer review of "DASH Dietary Pattern and Cardiometabolic Outcomes: An Umbrella Review of Systematic Reviews and Meta-Analyses"

_nutrients, 2019, doi:10.3390/nu11020338_

Round 1

Reviewer 1 Report

This manuscript presents a review of systematic reviews and meta-analyses regarding the DASH dietary pattern and cardiometabolic outcomes. The manuscript is comprehensive and well-written.

A few comments for improvement:

Lines 94-6 are confusing: ‘Public health policy and clinical practice guidelines are established with the use of systematic reviews and meta-analyses of controlled trials and prospective cohort studies, which are regarded as the highest levels of evidence.’ But yet GRADE criteria initially classify controlled trials as high-quality evidence and uncontrolled (even prospective) as low-quality?

Line 236: ‘The relationship, however, remains uncertain.’ Do the authors mean the risk, rather than relationship?

Lines 375-6: ‘The observed cardiovascular benefits may be attributable to the advice on a combination of the foods as part of the DASH dietary pattern’. The wording is unclear. I presume the authors mean that the observed benefits may be attributed to any/a combination of the foods within the DASH diet?

Please explain why the review start date was 2012.

Link implications section to the statements in the Introduction regarding updating EASD clinical practice guidelines. If this is the principal aim of the review, I believe it is important to separate out those studies including people with diabetes and present the evidence accordingly. The authors’ statement that there is no ‘biological reason to believe that the dietary pattern would behave differently in 484 people with diabetes’ is fair but unsubstantiated. Thus, I see no reason not to include studies with non-diabetic participants, but feel that the manuscript would be improved with the data presented separately.

Table 1 contains useful information, although it needs re-formatting for ease of reading.

Author Response

 Response to Reviewers:

Thank you very much for your review of our manuscript and for your helpful comments. Please see our responses in bold.

Reviewer #1:

This manuscript presents a review of systematic reviews and meta-analyses regarding the DASH dietary pattern and cardiometabolic outcomes. The manuscript is comprehensive and well-written.

A few comments for improvement:

Lines 94-6 are confusing: ‘Public health policy and clinical practice guidelines are established with the use of systematic reviews and meta-analyses of controlled trials and prospective cohort studies, which are regarded as the highest levels of evidence.’ But yet GRADE criteria initially classify controlled trials as high-quality evidence and uncontrolled (even prospective) as low-quality?

Thank you for highlighting this. We have revised the sentence to read “which are regarded as the best levels of evidence”.

Line 236: ‘The relationship, however, remains uncertain.’ Do the authors mean the risk, rather than relationship?

Thank you. Since the GRADE score was low for the effect of the DASH dietary pattern on diastolic blood pressure (DBP), the conclusion is that the relationship, or effect of the DASH dietary pattern on DBP remains uncertain. We have revised this sentence to clarify the conclusion based on the GRADE assessment.

Lines 375-6: ‘The observed cardiovascular benefits may be attributable to the advice on a combination of the foods as part of the DASH dietary pattern’. The wording is unclear. I presume the authors mean that the observed benefits may be attributed to any/a combination of the foods within the DASH diet?

Thank you for highlighting this. You are correct. We have revised our wording accordingly.

Please explain why the review start date was 2012.

Thank you. Our apologies for not being clear. The start date of our systematic search was not 2012, but the dates of inception of the databases. 2012 was the earliest census date of one of the included systematic reviews and meta-analyses. We have revised the wording.

Link implications section to the statements in the Introduction regarding updating EASD clinical practice guidelines. If this is the principal aim of the review, I believe it is important to separate out those studies including people with diabetes and present the evidence accordingly. The authors’ statement that there is no ‘biological reason to believe that the dietary pattern would behave differently in 484 people with diabetes’ is fair but unsubstantiated. Thus, I see no reason not to include studies with non-diabetic participants, but feel that the manuscript would be improved with the data presented separately.

Thank you. Although it would be ideal to have data separate for those with and without diabetes, since this is an umbrella review, we are summarizing the findings of the most recent systematic reviews and meta-analyses, which did not separate the data in this way. This is a point in our limitations, which we have now added to include discussion of the studies in diabetes.

Table 1 contains useful information, although it needs re-formatting for ease of reading.

Thank you. We have made some modifications to help with ease of reading Tables 1 and 2.

Reviewer 2 Report

The manuscript was well written and comprehensive. Figures 2 and 3 presented the reader with an nice summary of the most relevant information. Supplementary information was comprehensive. Below are a few recommendations for edits.

In the beginning of methods you indicate the review ended January 2019 but it was not until line 475 that the time period of the review (January 2012 to November 2018) was noted. Please include the time period in the methods.

Please edit the manuscript to reduce the number of "we" since we should not be used in scientific writing.

Line 113, it appears the "search of" is missing before systematic reviews.

Line 143 and Figure 1 There is an inconsistency in the total number of reports - text states 124 and figure 125 reports. Please edit for accuracy

Two spellings were used for fiber (fibre) please edit for consistency.

The authors noted nutrients in fruits and vegetables that are associated with decreased inflammation. A sentence could be added regarding flavonoids from these foods which are also known to decrease inflammation

In the introduction the authors state this review would be helpful in making recommendations regarding the DASH dietary pattern in clinical practice guidelines. After reading the discussion and conclusion it was not clear to me if a decision was made addressing this goal. If a recommendation was made please edit text.

Author Response

Response to Reviewers:

Thank you very much for your review of our manuscript and for your helpful comments. Please see our responses in bold.

Reviewer #2:

The manuscript was well written and comprehensive. Figures 2 and 3 presented the reader with an nice summary of the most relevant information. Supplementary information was comprehensive. Below are a few recommendations for edits.

In the beginning of methods you indicate the review ended January 2019 but it was not until line 475 that the time period of the review (January 2012 to November 2018) was noted. Please include the time period in the methods.

Thank you. To clarify, the literature search we conducted was from inception of the databases to January 3, 2019. In line 475, we are indicating what the range of census dates were for the included systematic reviews and meta-analyses. We have made some edits to help clarify this.

Please edit the manuscript to reduce the number of "we" since we should not be used in scientific writing.

Thank you for pointing this out. We have revised accordingly.

Line 113, it appears the "search of" is missing before systematic reviews.

Thank you. We have revised accordingly.

Line 143 and Figure 1 There is an inconsistency in the total number of reports - text states 124 and figure 125 reports. Please edit for accuracy

Thank you for highlighting this. We have made the correction to the text.

Two spellings were used for fiber (fibre) please edit for consistency.

Thank you. We have revised for fibre is spelled the same way throughout the manuscript.

The authors noted nutrients in fruits and vegetables that are associated with decreased inflammation. A sentence could be added regarding flavonoids from these foods which are also known to decrease inflammation.

Thank you. We have added flavonoids to the discussion around key nutrients and reduced inflammation.

In the introduction the authors state this review would be helpful in making recommendations regarding the DASH dietary pattern in clinical practice guidelines. After reading the discussion and conclusion it was not clear to me if a decision was made addressing this goal. If a recommendation was made please edit text.

Thank you. The DNSG of the EASD commissioned a series of systematic reviews and meta-analyses on dietary patterns, of which this is one. We are just to provide the best evidence on the DASH dietary pattern. All these systematic reviews and meta-analyses will be used by the DNSG of the EASD to make recommendations on dietary patterns and diabetes.
